# Influence of Cathodic Water Invigoration on the Emergence and Subsequent Growth of Controlled Deteriorated Pea and Pumpkin Seeds

**DOI:** 10.3390/plants9080955

**Published:** 2020-07-29

**Authors:** Kayode Fatokun, Richard P. Beckett, Boby Varghese, Jacques Cloete, Norman W. Pammenter

**Affiliations:** 1School of Life Sciences, University of KwaZulu-Natal, Westville Campus, Private Bag X54001, Durban 4000, South Africa; varghese@ukzn.ac.za (B.V.); pammente@ukzn.ac.za (N.W.P.); 2School of Life Sciences, University of KwaZulu-Natal Pietermaritzburg, Private Bag X01, Scottsville 3209, South Africa; rpbeckett@gmail.com; 3Openlab “Biomarker”, Kazan Federal University, 420008 Kazan, Republic of Tatarstan, Russia; 4Department of Mathematical Sciences, University of Zululand, Private Bag X1001, KwaDlangezwa 3886, South Africa; cloeteja@unizulu.ac.za

**Keywords:** antioxidant enzymes, cathodic water, controlled deterioration, invigoration, seedling emergence, viability

## Abstract

The quality of seeds in gene banks gradually deteriorates during long-term storage, which is probably, at least in part, a result of the progressive development of oxidative stress. Here, we report a greenhouse study that was carried out to test whether a novel approach of seed invigoration using priming with cathodic water (cathodic portion of an electrolysed calcium magnesium solution) could improve seedling emergence and growth in two deteriorated crop seeds. Fresh seeds of *Pisum sativum* and *Cucurbita pepo* were subjected to controlled deterioration to 50% viability at 14% seed moisture content (fresh weight basis), 40 °C and 100% relative humidity. The deteriorated seeds were thereafter primed with cathodic water, calcium magnesium solution and deionized water. In addition, to study the mechanism of the impacts of invigoration, the effects of such priming on the lipid peroxidation products malondialdehyde (MDA) and 4-hydroxynonenal (4-HNE) and on the reactive oxygen species (ROS) scavenging enzymes superoxide dismutase and catalase were also determined in the fresh and deteriorated seeds. All priming treatments improved seed emergence parameters, subsequent seedling photosynthesis and growth relative to the unprimed seeds. In general, cathodic water was most effective at invigorating deteriorated seeds. Analysis of the lipid peroxidation products and antioxidant enzyme activities in invigorated seeds provided support for the hypothesis that the effectiveness of cathodic water in invigoration of debilitated orthodox seeds in general and of pea and pumpkin seeds in particular derive from its ability to act as an antioxidant.

## 1. Introduction

Orthodox seeds can be effectively conserved in seeds banks but, even during careful storage, can eventually deteriorate in vigour and viability [1,2]. The deterioration of seeds in gene banks is of global concern, as it affects the long-term conservation of genetic diversity of both wild and agricultural plants [3,4] essential for future breeding programs. In the future, it will be necessary to produce varieties that perform well under future climate change scenarios, particularly in sub-Saharan Africa, where the effects of climate change are likely to be severe [5,6]. Deterioration of seeds can cause reduced or complete loss of seedling emergence on the field [7,8,9]. Furthermore, “hangover effects” of deterioration may be carried through to later growth stages of the plant. Seedlings of deteriorated seeds may display reduced photosynthesis and transpiration [10,11], slower growth and ultimately lower yields [12]. Thus, even if poor quality seeds germinate or emerge, the quality of plants generated from such seed is not guaranteed [7,10,11].

The “oxidative stress” model for seed deterioration suggests that deterioration is a result of the continued production of reactive oxygen species (ROS) such as superoxide (O_2_^−^), hydrogen peroxide (H_2_O_2_) and the hydroxyl radical (OH) [13,14]. These ROS damage many biomolecules and, in particular. cause cellular dysfunction through lipid peroxidation [13,14]. Furthermore, when the moisture content of seeds rises above approximately (c.) 14%, for example during early stages of imbibition, lipid peroxidation also occurs through the activities of the enzyme lipoxygenase [15]. This enzyme causes the release of cytotoxic lipid peroxidation products such as malondialdehyde (MDA) and 4-hydroxy-2-nonenal (4-HNE) [13]. Seeds possess housekeeping enzymes which ameliorate the cytotoxic effects of ROS and lipid peroxidation products on seeds, including superoxide dismutase (SOD), catalase (CAT) and ascorbate peroxidase (APX). In addition, they also contain nonenzymatic antioxidants such as ascorbic acid, glutathione, flavonoids and α-tocopherols [13,16]. As seeds age/deteriorate, ROS production increases, anti-oxidative mechanisms are overwhelmed as biomolecules become oxidized, and the result is poor seedling emergence on the field and poor subsequent seedling growth and productivity [17,18].

While seed companies and seed banks try to reduce deterioration, for example by storing seeds at low temperatures and low humidity, ways of improving or “invigorating” seeds that have already become partly deteriorated are being regularly investigated [8,17]. One approach is “seed priming”, first proposed by Reference [19], and involves hydrating seeds to a point that allows pre-germinative metabolism to start, but it is not high enough to allow actual radicle emergence (to avoid seed embryos becoming desiccation-sensitive). Seeds are then either re-dried until they return to their original dry weight for later planting [20,21,22,23] or are planted immediately [24]. Priming agents used for seed invigoration include water (hydro priming), inorganic salt (halo priming), solid matrices (matrix priming) and plant nutrients (nutrient seed priming) [23,25]. Where plant mineral nutrients are used for seed priming, besides ameliorating the impact of lipid peroxidation, the nutrients have been reported to provide a transient source of plant nutrition at the early stages of plant growth [25,26].

While the mechanism of seed priming is uncertain, considering that the oxidative stress model for seed ageing is associated with uncontrolled and unbalanced production of ROS, priming probably reduces the production of ROS or reduces their harmful effects. We therefore hypothesised that priming with solutions containing an external supply of antioxidants may be particularly effective in reinvigorating deteriorated seeds, leading to earlier and higher field emergence and more vigorous seedling growth. In our previous studies on recalcitrant species [27], cathodic water (CW) has shown itself to be remarkably efficacious in reducing ROS activity and in permitting the production of 70% viable seedlings (root and shoot) from axes of cryopreserved *Strychnos gerrardii*, which had never before been achieved [27]. Also, exposure of explants to cathodic water has achieved what seemed to be impossible, that is (viz.) callus production (for indirect morphogenesis) of an endangered plant species, and has resulted in plantlet formation from 100% of nodal explants of a heavy-metal accumulator [28]. Our earlier work on orthodox species [23,29] studied the effects of priming seeds with cathodic water; the results clearly showed that cathodic water can improve the germination of deteriorated seeds from a range of species. 

In the work presented here, we used *Pisum*
*sativum* and *Cucurbita*
*pepo* to further investigate the benefits of priming or reinvigorating controlled deteriorated seeds. Specifically, we tested the ability of priming with cathodic water to improve seedling emergence and subsequent seedling growth; unelectrolysed CaMg solution (nutrient priming) and deionised water (hydro priming) were investigated alongside cathodic water for comparison. In addition, to study the mechanism of invigoration, we tested the effects of priming on the lipid peroxidation and the activities of ROS scavenging enzymes in seeds at about the beginning of phase III of seed imbibition [30].

## 2. Materials and Methods

The study was conducted in a green house and in the laboratories of the School of Life Sciences, University of KwaZulu-Natal, Westville campus, Durban South Africa. Average temperature was 23.52 °C, and relative humidity was 67%. 

### 2.1. Controlled Deterioration of Seeds

The initial water contents of the seeds used was determined gravimetrically on a fresh weight basis as 11.7% for *P. sativum* and as 6.7% for pumpkin. Seed water contents were raised to 14% for both species in a vapour chamber, and the seeds were then subjected to controlled deterioration (CD; accelerated ageing) at 40 °C and 100% relative humidity (RH) in a digital oven (Series 2000, Scientific, Massachusetts, USA). Seeds were sampled from the oven at 4-day intervals, and germination was tested. CD was continued until complete loss of germination, and the time required for 50% inhibition of germination was estimated.

### 2.2. Preparation of Cathodic Water and Seed Priming

A solution of 0.5 μM CaCl_2_.2H_2_O and 0.5 mM MgCl_2_.6H_2_O [31] (CaMg solution) was autoclaved, 200 mL was decanted into each of the two glass beakers, and platinum electrodes were immersed in the solutions: the anode in one beaker and the cathode in another. The circuit was completed with an agar-based potassium chloride salt bridge (30% KCl), and the solution was electrolysed at 60V potential difference using a Bio-Rad PowerPacTM Basic (Bio-Rad, USA) power pack for 1 h at room temperature, yielding anodic (oxidizing) water at pH 2.4 and cathodic (reducing) water at pH 11.2 [27]. The anodic water was discarded, and the cathodic water was used for invigoration of seeds within 1 h.

The priming solutions used were cathodic water, un-electrolysed CaMg solution and deionized water. Seeds (50 seeds per treatment) were hydrated by placing them between 20 layers of single-ply paper towel (Twinsaver Wiper Roll), which was placed on aluminium foil. To prime the seeds, 50 mL of the solutions was poured onto these paper towels. The aluminium foil containing the seeds was placed in plastic pouches, and after 24 h, just before radicle emergence, the seeds were dried back to their original masses under ambient laboratory conditions for 7 days and kept at 4 °C in air-tight bottles until required.

### 2.3. Plant Management/Experimental Design

Potting mix and multi-feed fertilizer used in this study were purchased from Grovida, Durban, South Africa. In all, there were eight treatments (six seed-priming treatments and two controls). The seed-priming treatments were aged seeds primed with cathodic water (ASP.CW), aged seeds primed with CaMg solution (ASP.CM), aged seeds primed with deionized water (ASP.DW), fresh seed (unaged) primed with cathodic water (FSP.CW), fresh seeds primed with CaMg solution (FSP.CM), fresh seeds primed with deionized water (FSP.DW) and two controls. The first control comprised seeds that were controlled deteriorated (CD) and unprimed (ASC), and the second control comprised fresh seeds that were subjected neither to control deterioration nor to priming (FSC).

The plants were grown in 2 L pots containing 800 g of potting mix and were watered as required. After CD and priming, five seeds were planted in each pot, with four pots per treatment arranged in a completely randomized design. Four weeks after planting, plants were thinned to one plant per pot. Thinning was carried out by removing the weakest plants in the pots [32]. Plants were supplied with nutrients once every two weeks with Grovida water-soluble “multi-feed fertilizer” (1 g L^−1^) comprising N, P, K, S and Mg at 193, 83, 153, 6.1 and 4.6 g kg^−1^, respectively. The fertilizer also contained the micronutrients Zn, B, Mo, Fe, Mn and Cu at 700, 1054, 63, 751, 273 and 75 mg kg^−1^, respectively.

### 2.4. Seedling Emergence 

Seedling emergence counts were taken daily until no further change was observed. The following parameters were measured: first day of emergence (FDE), final emergence percentage (FEP), mean emergence time (MET), emergence index (EI) and uniformity of emergence (UE) [33,34,35].

### 2.5. Physiological Measurements

Leaf chlorophyll content was measured in the third, fourth and fifth leaves (counting from the terminal bud) across all treatments using a handheld, self-calibrating and non-destructive Soil Plant Analysis Development (SPAD chlorophyll meter (model SPAD-502; Minolta Corp., Ramsey, NJ). Three measurements were taken on each of the three leaves at 10 weeks of growth. Carbon fixation and the maximal efficiency of Photosynthesis II (PSII) (*F_V_/F_M_*) were measured with Li-Cor 6400 portable photosynthesis measuring system, fitted with a standard chamber and configured as an open system (Li-Cor, Lincoln, NE). Measurements were taken at 8 weeks after planting across all treatments and replicates. Instantaneous measurements of CO_2_ assimilation rates were carried out at a CO_2_ concentration of 400 ppm, a flow rate of 500 mL min^−1^, a temperature of 25 °C, and a light intensity of 1000 µmol m^−2^ s^−1^ between 11:00 am and 2:00 pm. Three measurements were taken per plant on the third, fourth and fifth leaves from the terminal bud. *F_V_/F_M_* was determined on the third leaf from the terminal bud 8 weeks after planting following dark adaptation for 40 min.

### 2.6. Harvesting and Plant Tissue Analyses

Shoot heights were measured after 12 weeks of growth, after which plants were harvested and separated into leaves, stems and roots. The lengths of the roots and stems were also measured, the leaves were counted, and their area was measured with a leaf area meter (CI-202 Area Meter, CID, Inc., Washington, USA). The plants parts (root, stem and leaves) were oven dried at 65 °C until constant mass, after which their biomass was determined. 

For plant tissue analyses, plants were grown in separate pots as contained in Section 2.3, except that no fertilizer was applied. After 4 weeks of growth, the leaves were harvested and oven dried to constant mass. The leaves were then ground to pass a 1 mm screen; 0.5 g was wet digested, filtered and analysed using an Inductively Coupled Plasma Atomic Emission Spectrometer (ICPAES). Similarly, treated seeds and the control were also analysed.

### 2.7. Determination of MDA and 4-HNE

In an additional experiment, fresh and deteriorated seeds were hydrated in the priming solutions as described above. Seeds were not dried, but at about the beginning of phase III of the seed hydration, 1 g was homogenised in 5 mL of 20% (*w/v)* trichloro acetic acid (TCA) consisting of 0.5% (*w/v*) thiobarbituric acid. The homogenate was then incubated for 30 min at 95 °C [36], placed in an ice bath for 10 min and then centrifuged at 10,000× *g* for 10 min. The absorbance of the supernatant was read at 600 nm using a PowerWave™ microplate spectrophotometer (BioTek Instruments, Inc, USA). The content of MDA was calculated spectrophotometrically using ε_532_ = 155 mM^−1^ cm^−1^ and was expressed as mmol g^−1^ fresh mass (FM).

The content of 4-HNE was also estimated at about the beginning of phase III of seed hydration. Seeds (1 g) were homogenised in 5 mL of cold borate buffer (0.2 M, pH 7.4) at 4 °C. The homogenate was then mixed with 10% (*w/v*) TCA and centrifuged at 12,000× *g* for 15 min. The supernatant obtained was thoroughly mixed with 2,4-dinitrophenyl hydrazine (1 mg mL^−1^ in 0.5 M HCl). The complex obtained was kept at ambient laboratory conditions for 2 h after and then extracted in hexane and evaporated under liquid nitrogen. The residue was dissolved in methanol at the concentration of 4-HNE calculated spectrophotometrically using ε_350_ = 13.8 mM^−1^ cm^−1^ [37], and results were expressed as mmol g^−1^ FM.

### 2.8. Quantification of Antioxidant Enzymes

The effects of priming solutions on the activities of SOD and CAT in seeds of fresh and deteriorated seeds of *C. pepo* were tested. Seeds were hydrated in the priming solution until about the beginning of phase III of seed imbibition. Extraction was done in phosphate buffer saline at pH 7.4 (1 g seed/5 mL buffer). The mixture was centrifuged for 10 min at 5000 rpm at 4 °C. The supernatant obtained was used to measure SOD and CAT activities. SOD activity was determined by measuring the percent inhibition of pyrogallol autooxidation by the enzyme at 420 nm [38]. Enzyme activity was expressed as units of SOD min^−1^ g^−1^ FM. CAT activity was measured using the method of Maehly and Chance [39] based on the breakdown of H_2_O_2_ (ε_240_ = 39.4 mM^−1^ cm^−1^), and activity was expressed as micro moles min^−1^ mg^−1^ protein. CAT activity was measured in all treated seeds and the two controls. Each of the 8 treatments was replicated 3 times. 

### 2.9. Statistical Analyses

The data collected were subjected to analysis of variance (ANOVA) using GenStat Release 12.1 (PC/Windows Vista) (VSN International Ltd., 2009). Means of the treatments were compared using Tukey Test at 5% least significant difference (LSD_0.05_). 

## 3. Results

### 3.1. Effect of Priming on Seedling Emergence and Growth

Seedling emergence was delayed as a result of controlled deterioration of seeds for 2 days in *P. sativum* and for 4 days in *C. pepo* (Table 1). Total seedling emergence, mean emergence time, emergence index and uniformity of emergence were also adversely affected by controlled deterioration when compared with the fresh unprimed seeds. In response to priming the controlled deteriorated seeds of *P. sativum* with cathodic water, a significant improvement of 33% occurred. The benefits of priming deteriorated seeds with CaMg solution and deionized water were smaller and did not differ significantly from unprimed seeds (Table 1).

With the exception of root length in *C. pepo*, controlled deterioration significantly (*p* < 0.05) reduced all growth parameters of both species (Table 2 and Table 3). Priming significantly improved most of the growth parameters in both control and deteriorated seeds; these effects were however much pronounced in the deteriorated seeds. Almost invariably, cathodic water was more effective than CaMg or deionized water. For example, in *P. sativum*, the total biomass of seedlings derived from aged seeds, cathodic water, CaMg water and deionized water increased yields from 0.8 g plant^−1^ to 2.9, 2.1 and 2.7 g plant^−1^, respectively. In the seedlings derived from fresh seeds, only cathodic water treatment increased total biomass significantly from 2.4 to 6.9 g plant^−1^ (Table 3).

### 3.2. Effect of Priming on Physiological Parameters

The seedlings produced from seeds derived from deteriorated seeds of both species contained significantly less chlorophyll than those derived from fresh seeds. The declines in chlorophyll content were 33.4% and 22.1% for *P. sativum* and *C. pepo*, respectively (Figure 1). In *P. sativum*, all priming treatments increased the chlorophyll content of the leaves of plants derived from deteriorated seeds to a similar extent. In *C. pepo*, only cathodic water was effective in increasing the chlorophyll content of the leaves of seedlings derived from deteriorated seeds. Priming had little effect on the chlorophyll content of seedlings derived from fresh seeds. 

For both species, controlled deterioration significantly reduced the maximal efficiency of PSII (*F_V_/F_M_*) of the resulting seedlings (Figure 1). All priming treatments restored values to those of the controls. By contrast, priming had little effect on fresh seed treatments. 

There was significant reductions of 50% (*P. sativum*) and 36.9% (*C. pepo*) in the rates of carbon fixation in the plants generated from fresh seeds compared to plants produced from deteriorated seedlings—ASC (Figure 2)—but priming with cathodic water increased the rates of fixation to values similar of those in seedlings derived from control seeds (FSC) in *P. sativum* and *C. pepo*. Priming with CaMg solution and deionized water increased rates of fixation, but the rates were not significantly higher than those in plants derived from unprimed seeds. In seedlings produced from fresh seeds, all priming treatments tended to slightly increase rates of fixation.

### 3.3. Effect of Priming on Lipid Peroxidation and Antioxidant Enzymes

Controlled deterioration significantly increased (*p* < 0.05) the content of both MDA and 4-HNE in *P. sativum* and *C. pepo* seeds (Figure 3). While the 4-HNE content increased by 1.74 in *P*. *sativum*, in *C*. *pepo*, there was a lower increase of 1.22-fold. Similarly, MDA content increased in the deteriorated seeds of *P. sativum* and of *C. pepo* by 1.69- and 1.51-folds, respectively (Figure 3). Cathodic water significantly reduced MDA and 4-HNE contents in controlled deteriorated seed treatments in both species. For MDA, the reductions were significant for cathodic and CaMg solution but not deionized water. For 4-HNE, the reductions were significant for all priming treatments in *P. sativum* and cathodic water was significantly more effective than either CaMg solution or deionized water. *In C. pepo*, only cathodic water significantly reduced 4-HNE levels. Priming had little effect on the levels of lipid peroxidation products in fresh seeds.

Controlled deterioration significantly reduced the activities of CAT and SOD in the seeds of *C. pepo* (Figure 4). For SOD, all priming treatments significantly increased SOD activity but cathodic water and CaMg solution were significantly more effective than deionized water. Generally, controlled deteriorated seed treatments benefited more that fresh seed treatments. While increases of 18.23%, 18.23% and 16.33% occurred in the fresh seeds treated with cathodic water, calcium magnesium solution and deionised water, respectively, much greater increases of 80.0%, 79.9% and 68.5% were obtained when similar treatments were applied to aged seeds (Figure 4). For CAT, only cathodic water and CaMg solution significantly increased enzyme activity and cathodic water was significantly more effective than CaMg solution.

### 3.4. Effect of Cathodic Water, Calcium Magnesium Solution and Deionized Water on Mineral Content of Seeds and Shoot of Pisum sativum and Cucurbita pepo

Priming the controlled deteriorated seeds of *P. sativum* with cathodic water and CaMg solution led to increase in the concentrations of magnesium in the seeds and of calcium in the shoot of *P. sativum* when compared with the unprimed controlled deteriorated (Figure 5). In *C. pepo,* there were neither significant changes in any of the treatments nor any significant elevation of Ca and Mg in any of the other *P. sativum* treatments (Figure 5).

## 4. Discussion

Cathodic protection as commonly understood is a process by which electrons generated at a cathode counteract oxidative corrosion. In the work presented here, a novel approach to cathodic protection was used, involving the reduced, cathodic fraction (“cathodic water”) of an electrolysed dilute solution of calcium and magnesium chloride [27] to invigorate controlled deteriorated seeds of *P. sativum* and *C. pepo*. Here, we compared the effects of priming with cathodic water with priming with un-electrolysed CaMg solution and deionized water. 

### 4.1. Effects of Priming on Emergence and Later Seedling Growth

Seedling emergence was delayed as a result of controlled deterioration of seeds in both species (Table 1). The delay in emergence may be linked to reduction in seed vigour and possibly also to the induction of secondary dormancy [40,41]. Delays in emergence as a result of loss of seed vigour have been reported in many plants including *Phaseolus vulgaris* L. [3], *Glycine max* [17] and *Zea mays* [41]. Generally, seed priming with cathodic water improved the emergence parameters in both species, with most of the improvements being statistically significant (Table 1). In particular, uniformity is an important parameter, as high uniformity has been reported to improve stand establishment, particularly important under suboptimal conditions such as drought, salinity and water stress. Uniformity of emergence due to priming has also been reported to supress weeds growth; for example, priming has been reported to reduce weed-induced yield loss in *Oryza sativa* by 10% [42]. While priming with CaMg solution or deionized water also generally improved the emergence parameters of deteriorated seeds, they were less effective than cathodic water and their effects were not always significant. In deteriorated seeds, cathodic water appears to have performed better than CaMg solution and deionized water although the values may not be statistically different in this study due to large variations. For instance, in *C. pepo,* emergence in ASP.CW was 12.5% greater than ASP.CM/ASP (Table 1). In terms of practical applications, such an increase is considered significant in terms of plant productivity [42] and may have a huge impact both in terms of recovery of germplasm and profitability (economic value/farm income). It is well known that priming can improve emergence in a variety of species (e.g., *Cicer arietinum* [9], *Zea mays* [41], *Triticum aestivum* [43] and *Oryza sativa* [44]; cathodic water appears to be superior to other commonly used priming solutions. 

Interestingly, in the present study, seeds were deteriorated to 50% emergence in petri dishes in the laboratory and, in pots in the greenhouse, emergence was greater than 50% in both species. The greater emergence may mean that the growth medium used was more favourable to the seeds than the germinating papers used for the germination study. The implication is that that laboratory germination data may not reflect emergence under field conditions.

While good germination is important, the subsequent growth of seedlings was reduced if they were derived from deteriorated seeds (Table 2 and Table 3). It is well known that “hangover effects” of deterioration may be carried through to later growth stages of the plant. Seedlings of deteriorated seeds may display reduced photosynthesis, slower growth and ultimately lower yields [5]. Thus, even though deteriorated seeds may germinate and emerge well, the resulting seedlings may have low “vigor”. Current models of seed deterioration suggest that, as seeds age, the accumulation of free radicals causes profound cellular damage, disrupting normal cell functions [17,43]. Lipid peroxidation impairs critical cellular function such as oxidative phosphorylation, reducing energy (ATP) production, resulting in slower cell division (mitosis) [12,45,46] and, ultimately, leading to a delay in emergence and plant growth [12,47]. 

In both species, controlled seed deterioration significantly reduced growth parameters such as root length and mass, stem length and mass, the number of leaves, leaf area and total biomass (Table 2 and Table 3). Invigoration of seeds of the test species with any of the priming solutions had some measure of improvement on root and stem length; number of leaves and leaf area; and the masses of the root, stem, leaf and shoot, as also the total biomass of the plants was derived from deteriorated seeds but was not significant (Table 2 and Table 3). However, cathodic water was almost invariably the best priming solution. For instance, not only did seed priming with cathodic water have significant impact on total biomass in both fresh and aged seeds treatments but also its improvement on total biomass was significantly better than what was obtainable from calcium magnesium solution and deionized water (with the exception of aged seeds in *P. sativum*). Also, priming was particularly effective in improving flower production. While deterioration completely supressed flower production in the seedlings of *C. pepo* (Table 2), priming with cathodic water produced *c*. two thirds the number of flowers as those from fresh seeds. Seedlings derived from deteriorated seeds primed with CaMg solution or deionized water produced less than one third of the flowers produced by seedlings derived from fresh seeds. Similar but less marked effects were observed in *P. sativum*. Reasons for the particularly strong effect of cathodic water priming on flower production remain unclear. The reduction in biomass of plant/plant parts as observed in this study might be due to the reduced nutrient uptake by the plants as a result of reduced root length/mass (Table 2 and Table 3). Our results are in agreement with the observations of many other researchers on a variety of species, for example Amanpour-Balaneji and Sedghi [7] (*Triticum aestivum*), Sreepriya and Girija [8] (*Sesamum indicum L*.), Wang et al [47] (*Oryza sativa*) and Umair et al [25] (*Zea mays*). Reductions in seedling growth a result of controlled deterioration of seeds were effectively alleviated by seed priming. It has been reported that seed priming enhances antioxidant activities, nutrient uptake, photosynthesis, rate of cell division and cell elongation [13,48,49,50,51]. 

### 4.2. Effects of Priming on Photosynthetic Parameters

The effects of deterioration and priming on the photosynthetic parameters measured here were consistent with the growth data; deterioration strongly reduced the efficiency of PSII (*F_V_/F_M_*) (Figure 1). Generally, priming deteriorated seeds improved the photochemical efficiencies. Lower *F_V_/F_M_* values in the seedlings derived from unprimed deteriorated seeds undoubtedly contributed to their poor growth (Table 2 and Table 3). Results from other photosynthetic parameters also support the view that improved photosynthesis in seedlings derived from primed seeds contributes to improved seedling growth. Deterioration reduced the chlorophyll content in all species (Figure 1). Cathodic water reversed the effects of deterioration, and although not significant in all instances, the improvements were generally better than those in plants primed with CaMg solution or deionized water. In both *C. pepo* and *P. sativum* seedlings produced from deteriorated seeds possessed significantly reduced rates of photosynthesis (Figure 2) compared with seedlings derived from fresh seeds. Priming, particularly with cathodic water increased the photosynthetic rates of seedlings derived from deteriorated seeds and, even to some extent, in seedlings derived from fresh seeds (Figure 2). While we did not study in detail the mechanisms whereby priming improved photosynthesis, the greater leaf area and leaf number in the primed seed treatments (Table 2) would have provided greater surface area for photosynthetic activities and subsequent growth of the plants [50,52,53]. The longer roots produced by cathodic water treated seeds may have also enhanced nutrient uptake, which ultimately would have improved on photosynthesis [48,50].

### 4.3. Effects of Priming on Lipid Peroxidation Products, ROS Scavenging Enzymes and Composition

In both species, cathodic water significantly reduced both MDA and 4-HNE levels in deteriorated seeds. Although the CaMg solution reduced the MDA level in *P. sativum* and *C. pepo*, it failed to significantly reduce the 4-HNE level in *C. pepo.* All priming solutions significantly reduced 4-HNE in *P. sativum*, and cathodic water was significantly better at reducing the levels of 4-HNE than CaMg solution and deionized water (Figure 3). In this study, MDA contents increased in the deteriorated seeds of *P. sativum* and *C. pepo* by 1.69 and 1.51 times, respectively, while the 4-HNE content increased by 1.74 times in *P. sativum* and 1.22-fold in *C. pepo* (Figure 3). The increases in both MDA and 4-HNE indicate that deterioration of seeds occur as a result of lipid peroxidation [15,17]. Although seeds have internal mechanisms to counteract the damaging effect of ROS, at high levels of stress, the internal protection of seeds may not be enough [15]. Priming with cathodic water significantly reduced the contents of MDA and 4-HNE in the seeds of the test species. All other treatments appear to have reduced MDA and 4-HNE contents in seeds, but some of the reductions were statistically insignificant. Reductions in lipid peroxidation products by cathodic water strongly support the view that priming with cathodic water boosts or augments the internal antioxidative system in the seeds [13,17].

Oxidative stress occurs as a result of overproduction of ROS leading to lipid peroxidation [13,17]. The major ROS implicated in damage to seeds are superoxide (O_2_^-^), hydrogen peroxide (H_2_O_2_) and the hydroxyl radical (OH), all of which would normally be quenched by the endogenous antioxidant system [54]. Damage results when the seeds’ internal antioxidant systems are inadequate to cope with ROS produced under conditions promoting oxidative stress such as controlled deterioration [13]. Controlled deterioration significantly reduced the activities of the ROS scavenging enzymes SOD and CAT in the seeds of both species, presumably as a result of repeated reactions with ROS (Figure 4). Priming increased enzyme activity, most likely as a result of reductions in ROS formation prolonging enzyme life but possibly in addition by upregulating enzyme synthesis/activities.

Consistent with earlier submissions that priming is more beneficial to deteriorated seeds when compared with fresh seed, priming-induced increases in SOD activities were higher in aged seeds compared with fresh seeds. For example, in *C. pepo*, while “increases” of 18.23%, 18.23% and 16.33% occurred in the fresh seeds treated with cathodic water, calcium magnesium solution and deionised water, respectively, much greater increases of 80.03%, 79.91% and 68.48% occurred when similar treatments were applied to aged seeds (Figure 4). The main motivation for the present study was that cathodic water may be a potent priming agent because of its powerful antioxidative capacity. By scavenging ROS, cathodic water should therefore ameliorate the impact of controlled deterioration on emergence and later growth of the plants. The results obtained provided some support for this hypothesis. 

Calcium and Mg are macronutrients required for plant growth; their presence in cathodic water but at 1 mM concentration in CaMg solution is short of the optimal soil levels of Ca and Mg required for plant growth [55]. However, the result of the plant tissue analyses indicated that the contributions of Ca and Mg as plant nutrients cannot be completely ignored [50,51] because priming led to increase in the concentration of magnesium in the seeds and of calcium in the shoot of *P. sativum* when compared with the unprimed controlled deteriorated seeds (Figure 5). It can therefore be suggested that the better performance of CaMg solution when compared with deionized water or control was due to the little contributions of Ca and Mg present in the treatment [50,51]. Since it is clear that the effect of Ca and Mg as plant nutrients to the growth of the test species is benign in most cases, it may be suggested that the main strength of cathodic water lies in its strong anti-oxidative properties, which may have helped in the control of ROS activities associated with seed deterioration, thereby alleviating oxidative stress damage in cathodic water-treated seeds [56,57,58] and probably reduction in the seed CD-induced hangover effects [10]. 

The strong antioxidant properties of cathodic water [58] facilitated the repair of controlled deteriorated seeds of the test species. However, there are interspecies differences in terms of the intensity of the sensitivity of each species. The biochemical assays and physiological data all supported the growth data and further lend credence to the strong reducing power of cathodic water. It has no doubt contributed to a deeper and fundamental understanding of the processes involved in cathodic protection in plant material, especially reinvigoration of *P. sativum* and *C. pepo* in particular and orthodox seeded species in general. Although priming is usually recommended for poor-quality seeds, this study has shown that freshly harvested seeds may also benefit from priming. 

## 5. Conclusions

The deterioration of seeds in seed banks is of global concern, as it affects the long-term conservation of genetic diversity of both wild species and agricultural species essential for future breeding programs. In the future, it will be necessary to produce varieties that perform well under future climate change scenarios, particularly in sub-Saharan Africa, where the effects of climate change are likely to be severe. Here, we show that priming can invigorate deteriorated seeds of two crop species. Priming improves seed emergence, and moreover, the effects of invigoration were carried forward into plant growth. While all priming solutions were capable of some measure of invigoration, priming with cathodic water was most effective. This may have been a result of the strong antioxidant capacity of cathodic water or other mechanisms of cathodic water actions not yet known. Either cathodic water exerts its beneficial effect in ways other than ameliorating oxidative stress or further work is needed to more accurately assess amelioration of oxidative stress by cathodic water, for example protein carbonyl levels, the glutathione redox couple or direct measurements of ROS levels in plant tissues. In conclusion, priming-deteriorated orthodox seeds with cathodic water can improve both the emergence and subsequent growth of *P. sativum* and *C. pepo*. While requiring extra time to prepare, the additional benefits of cathodic water suggest that it can be used as an effective tool in orthodox seed conservation.

## Figures and Tables

**Figure 1 plants-09-00955-f001:**
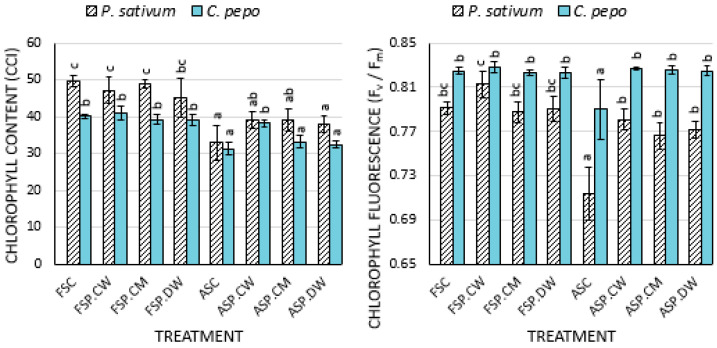
Effects of cathodic water, calcium magnesium solution and deionized water seed invigoration on the chlorophyll content and chlorophyll fluorescence of the leaves of *Pisum sativum* and *C. pepo* plants: Each treatment was replicated 4 times. Bars with different letters in each species are significantly different (*p* < 0.05). Chlorophyll fluorescence: Measurements were taken 1 time from 3 leaves per plant (*n* = 12/treatment). Chlorophyll content: measurements were taken 3 times per each leaf per plant (36 measurements/treatment).

**Figure 2 plants-09-00955-f002:**
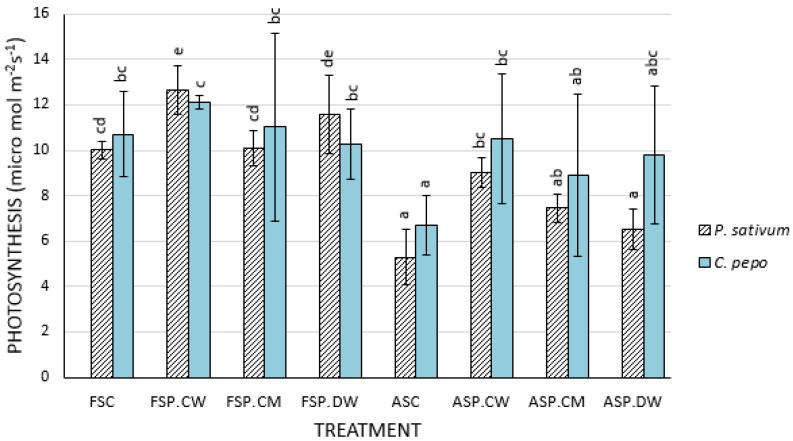
Effects of cathodic water seed invigoration on the photosynthesis of *Pisum sativum* and *Cucurbita pepo* plants: Bars with different letters in each species are significantly different (*p* < 0.05). Each bar represents the mean of 3 replicates; *n* = 12/treatment.

**Figure 3 plants-09-00955-f003:**
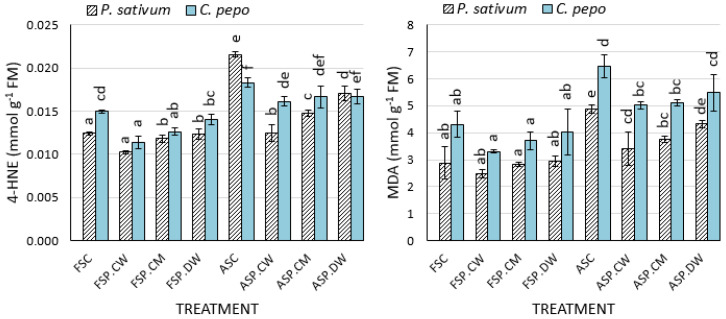
Effects of cathodic water, calcium magnesium solution and deionized water on the malondialdehyde (MDA) and 4-hydroxynonenal (4-HNE) content in the aged and primed seeds of *P. sativum* and *C. pepo*: Bars with different letters in a species are significantly different (*p* < 0.05). Each bar represents the mean of 4 replicates.

**Figure 4 plants-09-00955-f004:**
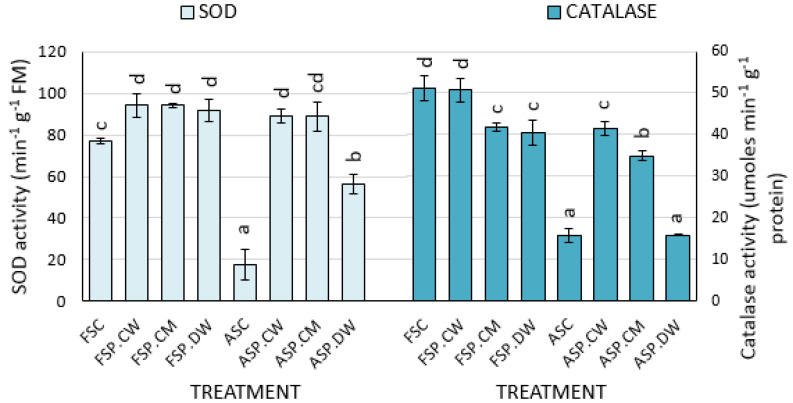
Effects of cathodic water, calcium magnesium solution and deionized water on the activities of superoxide dismutase (SOD) and catalase in *C. pepo*: Bars with different letters in a species are significantly different (*p* < 0.05). Each bar represents the mean of 4 replicates.

**Figure 5 plants-09-00955-f005:**
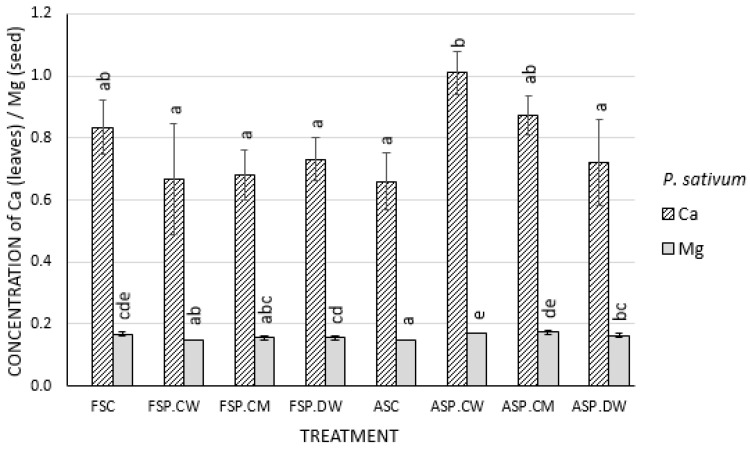
Concentration of magnesium in the seeds and calcium in the leaves of plants generated from primed controlled deterioration (CD) seeds of *Pisum sativum*: In both cases, Ca and Mg concentrations are higher in controlled deteriorated seeds primed with calcium magnesium solution and cathodic water treatments. Bar with different letters are significantly different (*p* < 0.05).

**Table 1 plants-09-00955-t001:** Effects of cathodic water, calcium magnesium solution and deionized water treatments on the emergence of *Pisum sativum* and *Cucurbita pepo.*

Emergence	FSC	FSP.CW	FSP.CM	FSP.DW	ASC	ASP.CW	ASP.CM	ASP.DW	LSD
***Pisum sativum***									
First day of emergence	4.3 ± 0.3 ^ab^	4.0 ± 0 ^a^	4.3 ± 0.2 ^ab^	4.5 ± 0.3 ^abc^	6.3 ± 0.2 ^d^	4.5 ± 0.3 ^abc^	5.3 ± 0.2 ^bcd^	5.5 ± 0.3 ^abc^	0.7 ± 0.4
Last day of emergence	6.3 ± 0.3 ^bc^	4.5 ± 0.3 ^a^	5.8 ± 0.2 ^b^	6.3 ± 0.2 ^bc^	7.5 ± 0.3 ^d^	6.3 ± 0.2 ^bc^	7.5 ± 0.3 ^d^	7.3 ± 0.2 ^cd^	0.8 ± 0.4
Emergence %	100.0 ± 0 ^c^	100.0 ± 0 ^c^	100.0 ± 0 ^c^	100.0 ± 0 ^c^	60.0 ± 0 ^a^	80 ± 8.2 ^b^	75 ± 5.0 ^ab^	70 ± 5.8 ^ab^	11.5 ± 5.6
Mean emergence time	4.1 ± 0.2 ^cd^	4.9 ± 0.04 ^d^	4.3 ± 0.2 ^cd^	3.9 ± 0.2 ^cd^	1.7 ± 0.2 ^a^	3.3 ± 0.4 ^bc^	2.6 ± 0.2 ^ab^	2.1 ± 0.2 ^a^	0.7 ± 0.3
Emergence index	3.0 ± 0.3 ^d^	4.3 ± 0.1 ^e^	3.2 ± 0.3 ^de^	2.8 ± 0.3 ^cd^	1.0 ± 0.2 ^a^	2.4 ± 0.3 ^bcd^	1.7 ± 0.2 ^abc^	1.3 ± 0.2 ^ab^	0.7 ± 0.3
Uniformity of emergence	0.14 ± 0.03 ^bc^	0.31 ± 0.01 ^d^	0.18 ± 0.04 ^c^	0.12 ± 0.02 ^abc^	0.03 ± 0.01 ^a^	0.09 ± 0.02 ^abc^	0.05 ± 0.01 ^ab^	0.04 ± 0.01 ^a^	0.06 ± 0.03
***Cucurbita pepo***									
First day of emergence	6.5 ± 0.5 ^b^	4.0 ± 0 ^a^	6.0 ± 0 ^ab^	6.0 ± 0 ^ab^	10.0 ± 0.8 ^c^	6.0 ± 0 ^ab^	6.5 ± 0.5 ^b^	7.0 ± 0.6 ^b^	1.2 ± 0.6
Last day of emergence	9.5 ± 0.5 ^b^	6.5 ± 0.50 ^a^	10.0 ± 0 ^b^	10.0 ± 0 ^b^	14.5 ± 0.5 ^d^	10.5 ± 0.5 ^bc^	11.0 ± 0.6 ^bc^	12.5 ± 0.5 ^cd^	1.3 ± 0.6
Emergence %	100.0 ± 0 ^c^	100.0 ± 0 ^c^	100.0 ± 0 ^c^	100.0 ± 0 ^c^	60.0 ± 0.0 ^a^	80.0 ± 8.2 ^b^	70.0 ± 5.8 ^ab^	70.0 ± 5.8 ^ab^	11.9 ± 5.8
Mean emergence time	4.3 ± 0.10 ^d^	4.8 ± 0.03 ^d^	4.3 ± 0.04 ^d^	4.12 ± 0.04 ^cd^	1.6 ± 0.10 ^a^	3.4 ± 0.30 ^bc^	2.8 ± 0.21 ^b^	2.6 ± 0.21 ^b^	0.5 ± 0.20
Emergence index	2.4 ± 0.10 ^d^	3.6 ± 0.09 ^e^	2.4 ± 0.06 ^d^	2.2 ± 0.08 ^d^	0.6 ± 0.10 ^a^	1.9 ± 0.20 ^cd^	1.4 ± 0.12 ^bc^	1.3 ± 0.16 ^b^	0.35 ± 0.17
Uniformity of emergence	0.015 ± 0.001 ^de^	0.021 ± 0.0004 ^f^	0.016 ± 0.0004 ^e^	0.014 ± 0.001 ^de^	0.006 ± 0.001 ^a^	0.013 ± 0.001 ^cd^	0.010 ± 0.001 ^bc^	0.009 ± 0.001 ^cd^	0.002 ± 0.001

There were eight treatments: FSC = fresh seeds that were neither aged nor primed; ASC = seeds that were aged but not primed; FSP.CW = fresh seeds that were not aged but primed with cathodic water; FSP.CM = fresh seeds that were not aged but primed with calcium magnesium solution; FSP.DW = fresh seeds that were not aged but primed with deionised water; ASP.CW = aged seeds that were prime with cathodic water; ASP.CM = aged seeds that were prime with calcium magnesium solution; and ASP.DW = aged seeds that were primed with deionised water. ANOVA was performed across treatments with the means of replicate separated at LSD_0.05_. Post hoc was done using Tukey’s test. Means along the same row with different letters were significantly different (*p* < 0.05, *n* = 32).

**Table 2 plants-09-00955-t002:** Effects of cathodic water, calcium magnesium solution and deionized water treatments on the root length, stem length, number of leaves, number of inflorescence and leaf areas of *Pisum sativum* and *Cucurbita pepo.*

Treatment	*Pisum sativum*					*Cucurbita pepo*				
	Root Length (cm)	Stem Length (cm)	Number of Leaves	Number of Inflorescence	Leaves Area (cm^2^)	Root Length (cm)	Stem Length (cm)	Number of Leaves	Number of Inflorescence	Leaves Area (cm^2^)
FSC	39.8 ± 0.7 ^c^	36.2 ± 0.5 ^e^	79.5 ± 1.0 ^c^	5.5 ± 0.3 ^cd^	266.8 ± 1.3 ^d^	30.5 ± 1.7 ^abc^	28.6 ± 3.0 ^de^	9 ± 0.4 ^bcd^	9.3 ± 0.6 ^c^	681.2 ± 8.3 ^c^
FSP.CW	45.5 ± 1.0 ^d^	45.5 ± 0.6 ^f^	97 ± 3.1 ^d^	9.5 ± 0.6 ^e^	309.4 ± 3.3 ^d^	51.6 ± 4.9 ^d^	42.8 ± 2.6 ^f^	11.3 ± 0.7 ^d^	10.8 ± 1.0 ^c^	768.4 ± 12.2 ^d^
FSP.CM	44.8 ± 0.9 ^d^	37.0 ± 0.9 ^e^	87.3 ± 1.3 ^c^	6.8 ± 0.5 ^d^	271.6 ± 12.8 ^d^	38.5 ± 2.4 ^bc^	21.4 ± 1.3 ^bcd^	9.5 ± 1.2 ^bcd^	9.5 ± 0.3 ^c^	680.9 ± 10.1 ^c^
FSP.DW	40.8 ± 1.6 ^cd^	36.6 ± 1.0 ^e^	85.3 ± 2.3 ^c^	6.5 ± 0.6 ^d^	269.4 ± 15.4 ^d^	38.8 ± 1.1 ^bc^	32.3 ± 1.1 ^e^	10.5 ± 0.3 ^cd^	6.5 ± 0.3 ^b^	701.2 ± 4.9 ^cd^
ASC	15.7 ± 0.8 ^a^	18.8 ± 0.4 ^a^	36.2 ± 1.2 ^a^	2.3 ± 0.2 ^a^	67.5 ± 1.1 ^a^	26.0 ± 1.6 ^a^	11.0 ± 0 ^a^	5.5 ± 0.2 ^a^	0.0 ± 0 ^a^	350.1 ± 4.3 ^a^
ASP.CW	37.3 ± 1.3 ^c^	31.1 ± 0.4 ^d^	52.5 ± 0.3 ^b^	4.5 ± 0.3 ^bc^	205.6 ± 3.8 ^c^	40.8 ± 0.7 ^cd^	27.1 ± 1.1 ^cde^	8.5 ± 0.3 ^bc^	6.5 ± 0.3 ^b^	645.9 ± 9.5 ^c^
ASP.CM	26.8 ± 0.7 ^b^	27.7 ± 0.6 ^c^	38.7 ± 1.6 ^a^	3.3 ± 0.2 ^ab^	143.3 ± 6.1 ^b^	35.8 ± 1.6 ^abc^	20.1 ± 2.2 ^bc^	7.5 ± 0.3 ^ab^	2.5 ± 0.3 ^a^	635.6 ± 11.1 ^c^
ASP.DW	23.0 ± 0.7 ^b^	22.8 ± 0.7 ^b^	51.4 ± 1.1 ^b^	2.8 ± 0.2 ^ab^	132.4 ± 1.7 ^b^	27.8 ± 2.14 ^ab^	13.4 ± 1.07 ^ab^	7.0 ± 0.41 ^ab^	2.5 ± 0.7 ^a^	457.4 ± 9.5 ^b^
LSD_0.05_	3.1 ± 1.5	2.0 ± 1.0	4.9 ± 2.3	1.2 ± 0.6	37.6 ± 2.2	6.7 ± 3.3	5.3 ± 2.5	1.7 ± 0.8	1.6 ± 0.8	48.2 ± 23.4

Seeds were subjected to controlled deterioration (ageing) at 100% relative humidity and 40 °C. There were eight treatments: FSC = fresh seeds that were neither aged nor primed; ASC = seeds that were aged but not primed; FSP.CW = fresh seeds that were not aged but primed with cathodic water; FSP.CM = fresh seeds that were not aged but primed with calcium magnesium solution; FSP.DW = fresh seeds that were not aged but primed with deionized water; ASP.CW = aged seeds that were prime with cathodic water; ASP.CM = aged seeds that were prime with calcium magnesium solution; and ASP.DW = aged seeds that were primed with deionised water. ANOVA was performed across treatments with the means of replicate separated at LSD_0.05_. Post hoc was done using Tukey’s test. Means along the same columns with different letters were significantly different (*p* < 0.05, *n* = 32). Means were sorted in ascending order.

**Table 3 plants-09-00955-t003:** Effect of cathodic water, calcium magnesium solution and deionized water treatment on the root mass, stem mass, leaves mass, shoot mass, total biomass and shoot/root ratio of *Pisum sativum* and *Cucurbita pepo*.

Dry Mass (g plant^−1^)	FSC	FSP.CW	FSP.CM	FSP.DW	ASC	ASP.CW	ASP.CM	ASP.DW	LSD_0.05_
***Pisum sativum***									
Root Mass	1.2 ± 0.01 ^bc^	1.6 ± 0.16 ^c^	1.6 ± 0.2 ^c^	1.2 ± 0.14 ^bc^	0.1 ± 0.01 ^a^	0.9 ± 0.07 ^bc^	0.6 ± 0.27 ^ab^	0.8 ± 0.19 ^ab^	0.5 ± 0.24
Stem Mass	1.0 ± 0.08 ^c^	1.5 ± 0.09 ^d^	1.1 ± 0.1 ^c^	1.1 ± 0.02 ^c^	0.3 ± 0.03 ^a^	0.7 ± 0.09 ^b^	0.5 ± 0.02 ^ab^	0.7 ± 0.03 ^b^	2.0 ± 0.09
Leaf Mass	1.1 ± 0.05 ^cd^	1.2 ± 0.09 ^d^	0.9 ± 0.12 ^c^	1.0 ± 0.04 ^cd^	0.2 ± 0.02 ^a^	0.6 ± 0.03 ^b^	0.5 ± 0.03 ^b^	0.5 ± 0.01 ^b^	0.2 ± 0.08
Shoot Mass	2.9 ± 0.11 ^d^	5.3 ± 0.08 ^e^	2.8 ± 0.15 ^d^	2.9 ± 0.04 ^d^	0.7 ± 0.03 ^a^	2.1 ± 0.12 ^c^	1.4 ± 0.03 ^b^	1.9 ± 0.03 ^c^	0.3 ± 0.12
Flower Mass	0.8 ± 0.03 ^b^	2.6 ± 0.11 ^c^	0.9 ± 0.06 ^b^	0.8 ± 0.01 ^b^	0.2 ± 0.01 ^a^	0.8 ± 0.05 ^b^	0.4 ± 0.01 ^a^	0.7 ± 0.01 ^b^	0.1 ± 0.07
Total Biomass	4.1 ± 0.09 ^c^	6.9 ± 0.16 ^d^	4.4 ± 0.34 ^c^	4.0 ± 0.16 ^c^	0.8 ± 0.04 ^a^	2.9 ± 0.10 ^b^	2.1 ± 0.30 ^b^	2.7 ± 0.23 ^b^	0.6 ± 0.29
Shoot/Root Ratio	2.4 ± 0.10 ^a^	3.4 ± 0.34 ^a^	1.9 ± 0.37 ^a^	2.6 ± 0.29 ^a^	14.4 ± 0.66 ^b^	2.4 ± 0.28 ^a^	5.1 ± 2.39 ^a^	2.9 ± 0.66 ^a^	2.7 ± 1.32
***Cucurbita pepo***									
Root Mass	1.0 ± 0.1 ^c^	1.8 ± 0.14 ^d^	1.0 ± 0.09 ^c^	1.1 ± 0.14 ^c^	0.3 ± 0.002 ^a^	0.8 ± 0.05 ^bc^	0.5 ± 0.02 ^ab^	0.4 ± 0.08 ^a^	0.26 ± 0.13
Stem Mass	1.6 ± 0.1 ^de^	2.2 ± 0.18 ^f^	2.0 ± 0.17 ^def^	2.1 ± 0.09 ^ef^	0.3 ± 0.02 ^a^	1.5 ± 0.03 ^cd^	1.0 ± 0.11 ^bc^	0.6 ± 0.09 ^ab^	0.32 ± 0.16
Leaf Mass	1.8 ± 0.2 ^bc^	2.1 ± 0.29 ^c^	2.0 ± 0.15 ^bc^	2.1 ± 0.15 ^c^	0.9 ± 0.01 ^a^	2.0 ± 0.17 ^bc^	1.4 ± 0.1 ^abc^	1.3 ± 0.16 ^abc^	0.51 ± 0.24
Shoot Mass	3.7 ± 0.3 ^cd^	4.8 ± 0.45 ^d^	4.1 ± 0.23 ^cd^	4.4 ± 0.10 ^cd^	1.2 ± 0.01 ^a^	3.6 ± 0.18 ^c^	2.5 ± 0.19 ^b^	2.0 ± 0.21 ^ab^	0.70 ± 0.34
Flower Mass	0.3 ± 0.02 ^de^	0.4 ± 0.03 ^e^	0.2 ± 0.03 ^c^	0.2 ± 0.01 ^cd^	0.0 ± 0 ^a^	0.2 ± 0.01 ^bc^	0.1 ± 0.01 ^ab^	0.08 ± 0.03 ^a^	0.07 ± 0.03
Total Biomass	4.7 ± 0.2 ^c^	6.5 ± 0.53 ^d^	5.1 ± 0.17 ^c^	5.5 ± 0.14 ^cd^	1.5 ± 0.01 ^a^	4.5 ± 0.22 ^c^	3.0 ± 0.21 ^b^	2.3 ± 0.28 ^ab^	0.80 ± 0.38
Shoot/Root Ratio	4.0 ± 0.5 ^ab^	2.70 ± 0.26 ^a^	4.4 ± 0.66 ^ab^	4.4 ± 0.6 ^ab^	4.2 ± 0.07 ^ab^	4.4 ± 0.17 ^ab^	5.0 ± 0.27 ^b^	5.7 ± 0.78 ^b^	1.40 ± 0.68

There were eight treatments: FSC = fresh seeds that were neither aged nor primed; ASC = seeds that were aged but not primed; FSP.CW = fresh seeds that were not aged but primed with cathodic water; FSP.CM = fresh seeds that were not aged but primed with calcium magnesium solution; FSP.DW = fresh seeds that were not aged but primed with deionised water; ASP.CW = aged seeds that were prime with cathodic water; ASP.CM = aged seeds that were prime with calcium magnesium solution; and ASP.DW = aged seeds that were primed with deionized water. ANOVA was performed across treatments with the means of replicate separated at LSD_0.05_. Post hoc was done using Tukey’s test. Means along the same row with different letters were significantly different (*p* < 0.05, *n* = 32).

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
