# Peer review of "Influence of Cathodic Water Invigoration on the Emergence and Subsequent Growth of Controlled Deteriorated Pea and Pumpkin Seeds"

_plants, 2020, doi:10.3390/plants9080955_

Round 1

Reviewer 1 Report

The manuscript entitled “Influence of cathodic water invigoration on the emergence and subsequent growth of controlled deteriorated pea and pumpkin seeds” describes the effect of priming on germination and early seedling growth from artificially aged pea and pumpkin seeds using cathodic water. 

Priming is obviously an important post-harvest technology for invigoration of seeds/seedlings, and the reported effects of cathodic water is pretty interesting. However, I find it required relatively large revision as some statement is not consistent with the data presented, some of the information is missing, and quite a few references are inappropriate (wrong citation).

  1. One general thing is that the result part only stated “improved significantly” in most of the cases. It would be better to describe how much, or as better as non-aged seeds, rather than repeating “significant” or “insignificant”. Also the discussion part is largely overlapped with results part. So it would be nice if discussed some other points, for example, this study showed cathodic water has a large potential for priming/ invigoration, then what to expect from the other cathodic water using other salts (I don’t know if it is possible or not, either, though…), is it going to be better? In another words, is there any room to improve the water even more? Or, unelectrolysed CaMg water in some cases better than water, how/why is that? 

  1. Line 45. I think “LOX” is usually used for “lipoxygenase”.  I might miss one, but I did not find “LOX” in the rest of the manuscript, therefore I suggest to remove “(LOX)”, just spell out as lipid oxidation products.

  1. Table 1, 2, 3. The table has LSD values and the values were presented with letters showing significance, but I find it easier to grasp the image of the variation among replicates if SE was added to the values. As long as I understood from the method part, each treatment has 4 replications, right?

  1. Line 95-98. Please refer to Table 3 in the text.

  1. Figure 1. Even though the detail was mentioned in the method part, it would be nice if a little more detail of the parameters were described in the legend – e.g. which leaf, when, replicate numbers, letters of significance etc, as figures should be self-explanatory. Also, the method part mentioned that measurement was done three times using different leaves, then are the data showing the mean of 12 measurements (from 4 independent plants)? Or 3 times per each leaf per plant (=36 measurements)?  Again, it would be nice if error bars are added.

  1. Figure 1. CCI in pea, FSP CW has “c” for significance, is it true? Is SD or SEM for FSP CW extremely small compared to the others?

  1. Line 142. It is mentioned that priming aged seed had a positive effect on CCI, but some are not significant. I do not understand it. Why is it not significant, although the letter for significance are different in Figure 1?  “a” for ASC, and “b” for all the primed ones?

  1. Line 144-147. This part is also confusing to me. How can you say cathodic water had the greatest improvement in both species when letters of significance are the same for all the priming treatment?  Am I missing something here?

  1. Figure 2. Again error bars and the more detailed legend is necessary.

  1. Line 160-163. This part is also confusing.  I thought Figure 2 shows the photosynthesis, or Carbon fixation, but this part describes transpiration citing Fig. 2.  Perhaps one figure is missing?

  1. Line 167. I think it is better to say lipid peroxidation rather than lipid oxidation.

  1. Figure 3. Again error bars and the more detailed legend would be nice and some letters of significance are not readable here.

  1. Line 171-173. I am not so sure about the greatest effect of cathodic water for 4HNE and of CaMg water for MDA. To me, either looking nearly the same or only in one species.

  1. Figure 4. Legend and error bars: same as above.

  1. Figure 4. The left Y-axis, SOD (%) is strange. It should be “SOD activity” or “Inhibition of auto-oxidation of pyrogallol.” However, the method part says it is expressed as units of SOD min-1 g-1 FM. Please correct it appropriately. The right axis should be catalase activity, not catalase. The unit for activity is umoles min -1 mg (or ug)-1 protein not mL.  Slash (/) was used here, but the other part are all min-1 or g-1, so better to be consistent throughout.

  1. Line 187- Section 2.4. It is again confusing here.  The text describes many different things such as different independent ions (e.g. Ca, Mg, K, Cu etc.) are affected in leaves or seeds, but the figure 5 only shows Mg in seeds and Ca in leaves. Method part did not tell seeds were analysed.  Also this figure only have one column, does it mean the data are two-species combined together?

Also, what does it mean by “concentration (%)” on y-axis?

  1. Line 236-241.  I agree that sometimes one cannot see exactly the same effect or response from petri dish experiment in the lab when trying with soil or in the field, but how many seeds were used per independent plate to select 50% viability in a preliminary experiment?  To my knowledge, typical germination test would be done with minimally 30 seeds, in many cases up to 100 seeds per single replicate, with minimally 3 replications. With this number relatively stable data would be obtained.  But in this study, one pot only had 5 seeds, then most likely 2 or 3 seeds germinate per pot, I guess. With this small number, one seed counts for 20 %, so 60% germination differentiating from 50% cannot be deeply discussed here anyway.  

  1. Line 246-247 mitosis is one of the class/type of cell division, so “mitosis which is essential for cell division” sounds strange.  The same for the following sentence.

  1. Line 260-261 None of the Ref 7, 8, 11, 25 describe about the species mentioned in the text. All wrong citations.

  1. Line 270-273. This sentence is repetitive.

  1. Line 275. “counteract the damaging effect of lipid peroxidation” sounds a bit strange. Lipid peroxidation means a damage of membrane, so rather plant has a protective system “try not to have/ try to reduce” the occurrence of lipid peroxidation.

  1. Line 298. “SOD may have helped in catalyzing……..into hydrogen peroxide and molecular hydrogen” sounds strange and is incorrect.  SOD’s activity is exactly catalyzing the dismutation of superoxide, so it is not “may have helped”, more like “it suggests/indicates”, and the products are hydrogen peroxide and oxygen, not hydrogen.  

  1. Line 344- . It is mentioned that “the little contributions of Ca and Mg present in the treatment”. If one monitor mineral content right after the treatment (meaning primed seeds), then perhaps seeds absorbed some minerals as they imbibed although the both Mg and Ca concentration of priming solution is quite low, but in the shoot later on, soil was already nutrient mixed, and even fertilizer was supplemented in the growth period, these fertilizer’s effect should be more prominent, as the amount or concentration is way larger compared with 0.5 uM or 0.5 mM in the priming solution.

  1. Line 79, and Line 380. “unelectrophorized”?  Is it “unelectrolysed” ?

  1. Line 367. It is mentioned that the seed moisture content was raised to 14%, but not mentioned how it is measured.

  1. Line 375. “salt-bridge” what kind of and how much salt in it?

  1. Line 381-384.   It is mentioned that 50 mL solution was used for priming, but it does not tell about the priming condition at all.  For example, how much seeds were used or what kind of case was used, or what was the moisture level the treated seeds during priming – the study used seeds of two different species which have completely different mass. 

  1. Line 431. I do not think reference 34 says about MDA determination.

  1. Line 450. Reference 36 does not explain anything about determination of enzymatic activities. It looked me a completely unrelated literature (not even a biology but sociology?)

Minor points

  1. Line 169, 174. Species names in italics.
  2. Line 373 and many other places. Please use an appropriate subscript for chemicals.
  3. Line 430. Please spell out TBA.
  4. Line 431. I do not think reference 34 says about MDA determination.
  5. I feel some of the sentences are repetitive and have a bit complicated structure – make it difficult to read through.

Author Response

 Dear Reviewer,

We sincerely appreciate your thorough review of our paper. Your contributions have no doubt improved on the quality of our paper.

We have to the best of our knowledge adhered to your suggestions.

Thanks a lot

Kayode FATOKUN

Reviewer 2 Report

This study is dealing with interesting process about seed reinvigorating.

Abstract

Could you explain briefly here what it is cathodic water?

Introduction

Can you add the references for phase II and phase III of seed imbibition?

More precise formulation of the questions is needed. What is your hypothesis? Are you expected that seed treated by cathodic water will be germinate quicker? Or with high %? The same for seedlings.

Result section

line 96: how the yield was measured?

line 97: what does it means this numbers? You were harvesting plants after 12 weeks of growth but this numbers are really low, they seem to be the weight of few day old seedlings.

section 2.1. The result are unclear and general, can you add proper results into this section?

Why did you display some result in a table and another in the picture? The figure legend is not a self explanatory, it is not clear what the letters a, b, c d, e above the bars mean. Why is in the Fig. 1 different species in separate parts and in the Fig. 2 and 3 mixing? In fig. 5 missing legend with explanation of what the color of bar indicate (instead of this, this information is in fig. label, could you move it for consistency with previous Figures?)

Discussion section

line 220: why are you mentioned specifically this two species (Glycine max. and Zea mays)? If you are writing that it was reported for many species it would be better mentioned specifically those working with species closely related to your species. Or is it reported only for monocots?

line 223: "with most of the improvements being statistically significant" you mean 4/5 significant parameters? Or 3/5? The second mentioned seems to me rather as half of parameters were significant.

line 250: shortcut CD was not used before

line 334: "leaves of leaves both"?

line 336: "[55](Kaiser 2015."?

line 348: shortcut CW was not used before

Materials and Methods section

line 365: GPS coordinates are not relevant for greenhouse condition

line 386:  information about temperature and RH is duplicated from 365

line 389: what does it mead "aged seeds"? How old were they? What was the relationship between aged seeds and fresh seeds? Were they collected from the same place?

line 396: Germinate all five seeds planted in one pot? I suppose not. Have you adjusted the amount of fertilizer and water to the number of plants in the pot?

line 397: How did you choose which plant stay in the pot after two weeks?

line 398: How much fertilizer did you add?

Be consistent in time unit marking (sometimes is used "hr" sometimes "h").

Did you used deionized or distill water? You mentioned both.

Author Response

Dear reviewer,

We appreciate your review and efforts were made to carry out your suggestions to the best of our knowledge. Your contributions have no doubt improve on the quality of our manuscript.

Detail responses to your query are attached

Thank you

Kayode FATOKUN

Round 2

Reviewer 1 Report

In this revised version, I see author’s effort to answer to my questions and to revise/correct several mistakes, which obviously made the manuscript better. 

However, I still see quite a few inconsistencies between the presented data and their statement/interpretation – majority was still left unrevised. Most likely I failed to convey the points of problem to the authors, because I did not pick up every single case.  To me, if the data is not significantly different, then it is not worth or not right to mention as “it increased” or “it decreased” – because it is simply not possible to say there is a difference by the treatment. If the data look quite different in the graph, but it is not significant due to the too large variations from replications, then perhaps one would like to mention such as “it looks different (or it looks there is an positive effect) in the graph, but it is not significant in this experiment” or “ the result was not conclusive due to the large variation”.  (These things are sometimes easily visible if the error bars are added, that is why I have asked in the previous version – however, it is just my preference and of course it is not indispensable since presented data has “significance letters”.)  The same way, what is the point to claim e.g. “increased 1.01-fold” (line 302)? Would the authors like to say 1% increase meaningful in this experiment? Even when it is not significant? Do they say that they would obtain the very similar data when they repeat the experiment again?  I simply suggest not to list figures (each data) for everything automatically, rather better to describe the points one wants to highlight.

Please state things based on your presented data. It is not a good idea to state collectively “increased in ALL treatments” when some of the result did not show significant difference.

I have asked in the previous review if some results were incorrect or perhaps missing, and good to have some corrected and also newly added supplementary figure in this version.  It is up to the authors if the added result kept included or not, but most importantly one should not mention about or conclude something out of the unpresented data in the study – either to add the data or to remove statement.  Still transpiration data was mentioned with referred to Figure 2, it should be supplemental data.  Some of the transpiration results do not show similar results when compared with carbon fixation, but again it was stated in general “in the same direction as that of carbon fixation” but it is not true at all. Please critically look at your own data and think how much/what you can tell.    

Also in the previous round, I suggested some different discussion topic in a wide context because discussion largely overlaps with the result part.  The response from the author to some example was “ it is beyond the focus of this study” – I did not ask to add this data. There are so many studies about priming effect in the field, and the “selling point” of this study should be the use of cathodic water. Therefore, I thought it is worth to discuss its future potential based on the results of this study or actual applied use or something like that.  It was just my easy suggestion, so there is no need to take this suggestion, however, how do you respond if somebody says “ Well it looks nice, but essentially on majority of the parameters tested here, it is possible to get only a slightly less or even similar level of effects simply using classic hydropriming or CaMg (nutrient) priming, then why should we need to use this special cathodic water? We need additional steps to prepare it.” I guess you can find plenty of points of discussion – e.g. what kind of cases is it likely most beneficial? Which parameter was the best out of it? Which parameter will not gain such a big benefit compared with classic priming? I think the number of inflorescences is quite good, as the authors have discussed. However, it would be better in the clearer comparison against the classical priming, not just saying cathodic water improved these, as this paper’s focus is comparison among the different priming.  Will the different species respond similar manner? At least two species used in this study did not show everything similar.  In the abstract/introduction, the conservation purpose/gene bank was mentioned at the beginning, then the target would not be crop’s shelf life, rather wide range of species with the unknown level of damages from ageing, also with limited number of seeds, I guess. Then what kind of parameters can be (or should be) explored further? etc.

I guess if these things clearly logically discussed in comparative manner, and not detouring by focusing on “traditional priming effects themselves” or “priming effects in crop seeds/agricultural benefit” (as the these are not the main theme of this study), then it would be much more consistent story.

I still feel I would not be able to repeat the priming with the description in the method part.

I think some sentences could be more concise and clearer.

Also some sentences have repeat, and there are some typographical errors.

And lastly it would be editorial formatting error, the beginning of introduction is not from the manuscript, I think.

Author Response

Dear reviewer,

We have responded to your suggestions and this time we made good efforts to adopt all your suggestions including showing the error bars in the graphs and tables. To me, it has been a learning experience without having to School fees. 

Thanks a lot,

Kayode 

Reviewer 2 Report

POINT 2: Introduction section: Can you add the references for phase II and phase III of seed imbibition?

RESPONSE 2: Phase II and phase III were not mentioned in the introduction section.

This is not true, phase III is mentioned in the introduction section (line 101).

POINT 14: line 365: GPS coordinates are not relevant for greenhouse condition

RESPONSE 14: It was included because we reported the average temperature and humidity ‘Average temperature was 23.52 ï‚°C and relative humidity: 67 % (GPS co-ordinates -29.817897, 30.942771)

I suppose that average temperature and relative humidity was measured in reported area and in your greenhouse probably differ. What is the point of stating these values?

POINT 18: line 397: How did you choose which plant stay in the pot after two weeks?

RESPONSE 18: The general rule for thinning plants was followed: removal of the weakest plant

Yes, this make sence and I suppose this choice. Could you add this explanation into methods?

Author Response

Dear Reviewer,

We have corrected the manuscript as you suggested.

Thanks for your comments which we consider to have contributed greatly to the improvement of our manuscript

Thanks

Kayode FATOKUN

Round 3

Reviewer 1 Report

This revised version shows such a great transformation as the logic flow is much clearer with some changes in the order of results, better description of results, and good discussion based on the obtained results. I am happy with this current version. Well done.

As I do not think more information will come even though I again ask you (I mentioned in the second review that I still don’t feel I could repeat the priming experiment based on the method described, but nothing was changed in this version.), and I will leave it as it is. But I just would like to tell you why I kept asking about the priming method. 

In agri-industry or seed technology area, it is quite important to control moisture content of the seeds during the priming.  As the method part only says you used how many layers of paper with 50 mL water, I cannot reproduce your experimental condition.  If you use thin paper such as Kimwipes, it could be like normal imbibition making the seeds completely wet, whereas thicker paper towel or filter paper, it could be the moisture limiting condition.  It also depends on the size of those paper - small paper with more available water, large paper more water limited. Either you mention the seed moisture content or explain a bit more about the paper (such as size and thickness or perhaps weight?- so that one can estimate how many layers of the other paper required or something).  If you use rather well known one such as Kimwipes (guess you can see it everywhere), then you can name it, so we all know what kind of paper.  So please be careful in the future experiments.

Author Response

Dear reviewer 1,

Like I have said in my previous notes, I consider it a privilege that you are involved in the review of our m/s. Your criticism/ contributions really challenged us and have ultimately lead to the improved version of the m/s that we now have. Thanks a lot

Kayode
